# The pluripotency factor NANOG contributes to mesenchymal plasticity and is predictive for outcome in esophageal adenocarcinoma

Amber P. van der Zalm [1,2,3], Mark P. G. Dings [1,3,4], Paul Manoukian [1,3,4], Hannah Boersma[1], Reimer Janssen[1], Peter Bailey [5], Jan Koster [1,2], Danny Zwijnenburg[1,2], Richard Volckmann[1,2], Sanne Bootsma[1,3,4], Cynthia Waasdorp[1,3,4], Monique van Mourik[2,3], Dionne Blangé[1,2,3], Tom van den Ende[1,2,3], César I. Oyarce[1,2], Sarah Derks [4,6], Aafke Creemers[1], Eva A. Ebbing[1], Gerrit K. Hooijer [7], Sybren L. Meijer[7], Mark I. van Berge Henegouwen [2,8], Jan Paul Medema[1,3,4], Hanneke W. M. van Laarhoven[2,3,9] & Maarten F. Bijlsma[1,3,4,9] ✉

## Abstract

**Background** Despite the advent of neoadjuvant chemoradiotherapy (CRT), overall survival rates of esophageal adenocarcinoma (EAC) remain low. A readily induced mesenchymal transition of EAC cells contributes to resistance to CRT.
**Methods** In this study, we aimed to chart the heterogeneity in cell state transition after CRT and to identify its underpinnings. A panel of 12 esophageal cultures were treated with CRT and ranked by their relative epithelial-mesenchymal plasticity. RNA-sequencing was performed on 100 pre-treatment biopsies. After RNA-sequencing, Ridge regression analysis was applied to correlate gene expression to ranked plasticity, and models were developed to predict mesenchymal transitions in patients. Plasticity score predictions of the three highest significant predictive models were projected on the pre-treatment biopsies and related to clinical outcome data. Motif enrichment analysis of the genes associated with all three models was performed.
**Results** This study reveals *NANOG* as the key associated transcription factor predicting mesenchymal plasticity in EAC. Expression of *NANOG* in pre-treatment biopsies is highly associated with poor response to neoadjuvant chemoradiation, the occurrence of recurrences, and median overall survival difference in EAC patients (>48 months). Perturbation of NANOG reduces plasticity and resensitizes cell lines, organoid cultures, and patient-derived in vivo grafts.
**Conclusions** In conclusion, NANOG is a key transcription factor in mesenchymal plasticity in EAC and a promising predictive marker for outcome.

## Plain Language Summary

Esophageal cancer is the sixth most common cause of cancer-related death worldwide. Although chemotherapy combined with radiotherapy (chemoradiotherapy) followed by surgery has improved survival, tumor recurrence and metastatic disease (that has spread to other parts of the body) are often observed after several months. In this study, we assessed the effect of chemoradiotherapy on esophageal cells in the lab to predict the effect in patients with esophageal cancer. To investigate this, genes were assessed from 12 different cell lines and 100 patient tissues. We revealed that levels of one of the genes, *NANOG*, associates with poor response in patients. NANOG could be a promising marker to predict outcome in patients with esophageal cancer. This knowledge might help clinicians to treat patients with esophageal cancer appropriately, or may lead to new or optimized treatments.

[1]Amsterdam UMC location University of Amsterdam, Center for Experimental and Molecular Medicine, Laboratory of Experimental Oncology and Radiobiology, Amsterdam, The Netherlands. [2]Cancer Center Amsterdam, Cancer Biology, Amsterdam, The Netherlands. [3]Amsterdam UMC location University of Amsterdam, Department of Medical Oncology, Amsterdam, the Netherlands. [4]Oncode Institute, Amsterdam, Netherlands. [5]School of Cancer Sciences, University of Glasgow, Glasgow, UK. [6]Amsterdam UMC, Vrije Universiteit Amsterdam, Department of Medical Oncology, Cancer Center Amsterdam, Amsterdam, the Netherlands. [7]Amsterdam UMC location University of Amsterdam, Department of Pathology, Amsterdam, the Netherlands. [8]Amsterdam UMC location University of Amsterdam, Department of Surgery, Amsterdam, the Netherlands. [9]These authors jointly supervised this work: Hanneke W.M. van Laarhoven, Maarten F. Bijlsma. ✉e-mail: m.f.bijlsma@amsterdamumc.nl

Esophageal cancer is the sixth most common cause of cancer-related death worldwide and can be classified according to the histological subtypes esophageal adenocarcinoma (EAC) and esophageal squamous cell carcinoma (ESCC)[1]. Although neoadjuvant chemoradiation (CRT) followed by surgery (CROSS) has significantly improved ten-year survival rates for both subtypes, resistance mechanisms are at play and tumor recurrence and metastatic disease are often observed after several months[2,3]. These mechanisms of therapy resistance are poorly understood.

Several factors contribute to resistance against therapy, including tumor microenvironmental constituents, hypoxia, and senescence[4-7]. A consistent factor contributing to therapy resistance in EAC is the high plasticity of these cancer cells and the resultant readiness to undergo epithelial-to-mesenchymal transition (EMT)[8]. Through mesenchymal transition, EAC cells lose their epithelial morphology, become more mesenchymal and motile, but also resistant to therapy[8,9]. Paradoxically, we observed that high therapeutic pressure of multimodality treatments such as CRT, carry a particular risk of inducing mesenchymal transition[10].

Mesenchymal transitions are mediated by the activity of several transcription factors from members of the Snail, Twist, and Zeb families that contribute to cytoskeletal and morphological changes[11,12]. In addition, the mesenchymal transitions mediated by these transcription factors associate with the occurrence of stem cell-like populations[13]. This stemness likely contributes to therapy resistance and cancer recurrence and progression[14]. However, whether stemness is a direct result of mesenchymal transition, or rather a requirement for plasticity that allows the induction of mesenchymal cell states, is not fully understood. The aim of this study was to chart and interrogate the heterogeneity in CRT-induced mesenchymal transitions, to identify its mechanistic requirements and develop biomarkers to predict its occurrence. In this study, we provide novel insights in the mechanisms that explain the high plasticity of EAC cells and find that *NANOG* expression in pre-treatment EAC biopsies is highly predictive for response to therapy and resultant patient outcome.

## Methods

### Ethical approval
All patient material, primary cell lines, organoid cultures, and clinical data were collected with written informed consent under ethical approval by the Amsterdam UMC IRB (Medisch Ethische Toetsings Commissie) METC 2013_241. Baseline characteristics are stated in Supplemental Table S1. Approval was obtained for publication of summarized clinical variables that cannot identify individual patients.

### Establishment of EAC cell cultures
Primary EAC cell lines were previously established from resected patient material as described before[15]. See Supplemental Table S2 for characteristics. Primary cell lines were obtained and established in agreement with pertinent legislation, Declaration of Helsinki, and patient's informed consent. Primary EAC cell lines 007B and 031 M were cultured in Advanced DMEM/F12 (Gibco) supplemented with N2 (5 ml; Invitrogen), HEPES (5 mM; Life Technologies), D-glucose (0.15% v/v; Sigma-Aldrich), β-mercaptoethanol (100 μM; Sigma-Aldrich), Insulin (10 μg/ml; Sigma-Aldrich), Heparin (2 μg/ml; Sigma-Aldrich), and 1:1000 Trace elements B and C (Fisher Scientific)[16]. Primary cell lines 058B, 081R, and 289B were sorted for the tumor surface marker EpCAM to obtain a pure tumor cell population. 058B, 081R, and 289B were cultured in DMEM (Gibco) supplemented with FCS (10% v/v), L-Glutamine (2 mM; Sigma-Aldrich), penicillin and streptomycin (500 μg/ml). Publicly available EAC cell lines Flo1 (RRID: CVCL_2045), OE19 (RRID: CVCL_1622) and OE33 (RRID: CVCL_1622; ATCC, Manassas, VA) were cultured in RPMI, supplemented with FCS (8% v/v), L-Glutamine (2 mM) and penicillin and streptomycin (100 units/ml; all from Lonza, Basel, Switzerland). All cell lines were checked for mycoplasma each month.

In vitro chemoradiation protocol, In vitro modeling of the CROSS regimen was mimicked by radiation, paclitaxel, and carboplatin treatment as described previously[10]. Carboplatin and paclitaxel used for esophageal patients were purchased from the pharmacy of the Amsterdam UMC. Therapy scheme was according to the following sequence: Day 0, plating cells, Day 1–4 carboplatin at 20 μM and paclitaxel at 0.5 nM combined with 1 Gy radiation; day 5–6, no therapy. This cycle was repeated on day 7 until day 14 (Fig. 1a).

Quantitative real-time PCR RNA of was extracted using the NucleoSpin RNA kit (Bioké Macherey-Nagel, Düren, Germany). cDNA was synthesized using Superscript III (Invitrogen) and random primers (Invitrogen). Real-time quantitative RT-PCR analysis was performed using SYBR green (Roche) on a Lightcycler 480 II (Roche). Relative expression was calculated using the comparative threshold cycle (Cp) and normalized to *GAPDH* or *RPS18* as a reference gene. The primer sequences used for knockdown validation are shown below (Supplemental Table S3). Flow cytometry Cells were harvested using trypsin-EDTA (Lonza) and washed in FACS buffer (1% FCS in PBS). Cells were stained for 30 min at 4 °C with the following antibodies diluted in FACS buffer; anti-human CD324 (E-Cadherin, 1:200, Cat. No: 324105, BioLegend), anti-human CD326 (EpCAM, 1:200, Cat. No: 324243, BioLegend), anti-human CD184 (CXCR4,1:200, Cat. No: 306515, BioLegend), anti-human CD325 (N-Cadherin, 1:200, Cat. No: 350811, BioLegend), and anti-human CD29 (1:200, Cat. No: 303014, BioLegend). Intracellular epitopes were targeted using permeabilization buffer (BD Biosciences, San Jose, CA). Data were analyzed using FlowJo 10 (Tree Star, Ashland, OR). Geometric mean fluorescence (gMFI) intensity in the relevant channel was calculated by correcting for isotype control, yielding the ΔgMFI. For gating strategy, please see ref. 17.

### Immunofluorescent staining
Cells were grown on round coverslips. Using 4% paraformaldehyde and 1% Triton X-100 cells were fixed and permeabilized. Blocking of cells was done with Dako REAL peroxidase blocking solution (Agilent Technologies, CA) for 15 min and antibodies were diluted in BrightDiluent green (Immunologic, NL). Cells were incubated overnight at 4 °C with the following primary antibodies; mouse anti-vimentin (Santa Cruz, sc-73259, 1:300), rabbit anti-E-cadherin (Abcam, Ab40772, 1:300) and rabbit anti-laminin (Thermo Fisher, PA5-22901, 1:200). Cells were incubated for 1 h at room temperature with the following secondary antibodies; Alexa Fluor 448 anti-rabbit IgG1 (H + L, Invitrogen, A11008, 1:400) and Alexa Fluor 546 anti-mouse IgG (H + L, Invitrogen, A11030, 1:400). Actin staining was performed using ActinRed 555 Readyprobe (Thermo Fisher, R37112) and nuclear staining was with DAPI (Sigma Aldrich, D9541-5MG, 1:5000). Images were obtained on a SP8-X-DLS Confocal microscope (Leica).

### Patient participants
Eligible patients were ≥18 years with pathologically confirmed EAC. Gastroesophageal junction tumors were eligible if the bulk of the tumor was located in the distal esophagus or on the gastroesophageal junction. All patients provided written, informed voluntary consent for study participation. This study was conducted in accordance with the Declaration of Helsinki and the international standards of good clinical practice. Screening exclusion criteria were: <18 years of age and another active malignancy interfering with the prognosis of esophageal adenocarcinoma.

### Patient biopsy processing
Snap frozen esophageal tumor biopsy samples were collected in the Amsterdam UMC between 22 May 2013 till 1 June 2020 with ethical approval (BiOES; METC 2013_241). With a cryostat 20 μm slices of snap-frozen samples of biopsy tissue were cut. A representative slice of 5 μm in the center of the tissue was collected for subsequent H&E staining. An experienced pathologist of the Amsterdam UMC determined tumor percentage (SLM). Of 139 assessed esophageal biopsies, median tumor cellularity was 45% and 35%, respectively. Total RNA was isolated using the AllPrep DNA/RNA/miRNA universal kit (Qiagen, Hilden, Germany) according to the manufacturer's protocol. RNA was eluted in 30 μl RNAse-free water. NanoDrop (Thermo Fisher, Waltham, MA) was used to measure RNA

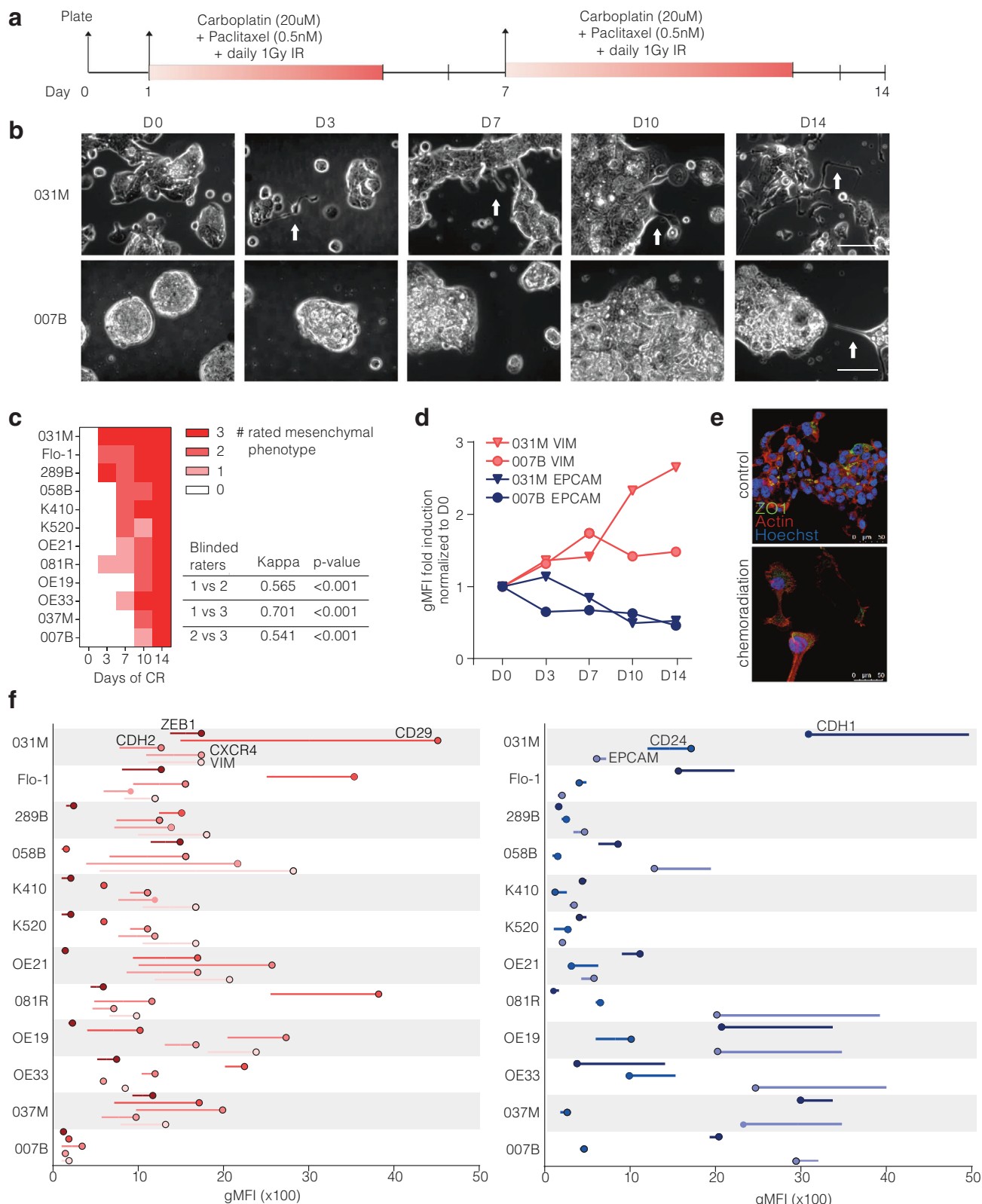

concentration. Samples were sent for RNA sequencing in case RNA concentration was above 20 ng/µL.

## RNA-sequencing

Cell lines and primary cultures in duplicate ($N = 24$) and 100 biopsies were processed for RNA-sequencing. Library preparation was performed using

Total RNA library prep RiboErase (Roche, Basel, Switzerland). Samples were sequenced in three batches on an Illumina HiSeq4000 with single 50 bp reads and 100 million reads per sample. All sequencing data were quality-controlled using FastQC42 and found to be of high quality. RNA-Seq reads were aligned to the human reference genome (NCBI37/hg19) using using STAR v2.7.1 and annotated with Gencode v32, retaining only uniquely

**Fig. 1 | Esophageal cell lines show heterogeneity for plasticity after chemoradiation. a** Diagram depicting the schedule of chemoradiation therapy (CRT). Day 0, plating cells, Day 1-4 carboplatin at 20 μM and paclitaxel at 0.5 nM combined with 1 Gy radiation; day 5–6, no therapy. This cycle was repeated on days 7–14. **b** Phase-contrast images taken over time during CRT treatment. Mesenchymal sprouting of cells indicated by white arrows. Scale bar indicates 70 μm. **c** Scoring of mesenchymal phenotype by three independent observers on blinded phase-contrast pictures. Inter-observer variability tested by Cohen's Kappa. **d** 007B and 031 M cells were exposed to CRT and harvested at indicated timepoints and stained for EpCAM and VIM. Data shown is geometric Mean Fluorescent Intensity (gMFI) of three technical replicates corrected to isotype control, normalized to day zero without treatment (D0). **e** 031 M cells were plated for microscopy, treated with CRT for 14 days and processed for immunofluorescence for ZO1 (green), actin (red), and nuclei (DAPI; blue). Magnifications and laser settings were identical between all images. Scale bars are 50 μm. **f** Flow cytometry analysis of mesenchymal markers and epithelial markers in the complete cell line panel, ranked for morphology-based propensity to mesenchymal plasticity. All cell lines and markers were measured at the same time point. Shown is gMFI of biological replicates of untreated samples (start of line; no dot) and CRT (dot).

mapped reads. The resulting gene expression profiles were converted into DESeq2_vst values using DESeq2 and log2-transformed. Non-biological batch effects were examined using PCA, and RUVg corrections were applied. Subsequent analyses were done on the batch-corrected dataset. Data were log2 transformed after alignment and normalization. Data were uploaded and analyzed in the R2: Genomics Analysis and Visualization Platform, or analyzed in R.

### Ridge regression analysis
Marker expression levels (delta gMFI) and in vitro generated morphological assessment scores were scaled and centered. To minimize non-informative genes, we selected the top 33% most expressed genes that are both expressed in tumor samples and cell lines. From this set, the top 5000 most variable genes were maintained[18]. Subsequently, for each phenotype an optimal lambda was calculated by using alpha=0 (ridge regression), Nfold = 8 (Leave-One-Out Cross validation)[19]. The optimal lambda was then used for the final model used to predict mesenchymal fates in the cell lines and patient samples.

### Imaging-based proliferation assay
Proliferation of cells on treatment was determined using an IncuCyte S3 (Sartorius). Live cells were detected before, during and after treatment. Phase contrast images were obtained, and confluence was analyzed by defining a confluence mask with a segmentation adjustment of 1 for all EAC cell lines and conditions. Confluence was then normalized for seeding density before treatment.

### Mouse experiments
Animal work procedures were approved by the animal experimental committee of the institute according to Dutch law and performed in accordance with ethical and procedural guidelines established by the Amsterdam UMC, location AMC and Dutch legislation. Ethical approval number was AVD1180020171672. NOD.Cg-Prkdcscid Il2rgtm1Wjl / Szj (NSG) mice were bred in-house. Animals were kept at room temperature in a DM2/ML2 animal facility with 4-6 animals per cage and were specific pathogen-free. From 12 weeks of age, 48 mice were included in the experiment, and subcutaneously injected in the right hind limb. $1 \times 10^5$ cells were injected in a volume of 100 μl with 50% medium and 50% Matrigel. After three weeks, with a a tumor size of approximately 100mm$^3$, 48 mice were blindly randomized to treatment groups, with 6 mice per group. Males and females were equally distributed over treatment conditions. Mice were daily treated for 2 weeks with 2 Gy radiation, Niclosamide or vehicle, simultaneously to minimize potential cofounders. After treatment, mice were monitored 3 times/week. All experiments ended for individual mice (determined a priori) either when the total tumor volume exceeded 500mm$^3$, when the tumor showed ulceration, in case of serious clinical illness, when the tumor growth blocked the movement of the mouse, or when the experiment was stopped 100 days after injection. Tumor size was assessed as outcome measure.

### Statistical analysis
Statistical analysis was done using GraphPad Prism version 9.3.1, Genomics Analysis and Visualization Platform R2 or R. Statistical tests are indicated in legends, and were considered significant $p < 0.05$ and indicated with;

$*p < 0.05$, $**p < 0.01$, $***p < 0.001$, $****p < 0.00001$. Error bars show the SD of the mean. Spearman correlation was determined with $p < 0.05$ considered significant. The Kaplan–Meier method was used to assess OS along with the log-rank test for statistical significance in R2 (patient data) or GraphPad (experimental data).

For the analysis of clinical variables pertaining to therapy response, the following selections were applied: For Fig. 2d, EAC and ESCC patients are included that received CROSS-only ($n = 44$; left panel). For Fig. 3a, EAC patients were included that received a chemoradiation-based therapy followed by resection (necessary to determine response to CRT). This includes CROSS also with add on therapies (with Trastuzumab and Pertuzumab, and with Nivolumab). These total 47. All these patients were treated with curative intent with no indication of distant metastasis at this point. For Fig. 3b, c (clinical outcomes) we also included patients that received definitive CRT and are not resected. All patients included in the analysis in Fig. 3 received some sort of CRT, and those who did not were excluded. These total 55. Note that for some samples variables are unavailable. For instance, Mandard scores are missing for 4 patients (leaving 43 patients), recurrence data is missing for 1 (50 left for analysis), and survival data 4 (55 left).

### Reporting summary
Further information on research design is available in the Nature Portfolio Reporting Summary linked to this article.

## Results
### EAC cultures harbor heterogeneous propensities for mesenchymal transition in response to chemoradiation
Mesenchymal cell state transitions in response to chemoradiotherapy (CRT) were studied in 6 publicly available cell lines, and 6 primary lines established from patient material using methods previously reported[15]. Together, these comprised 8 adenocarcinoma and 4 squamous cell carcinoma cultures. Supplemental Table S2 describes characteristics of the cell line panel including the clinical variables of the patients from whom the cell lines were derived. The panel was treated with an in vitro approximation of the CROSS regimen (Fig. 1a)[10]. During treatment, the onset of mesenchymal morphologies was assessed by microscopy (Fig. 1b). The timing of appearance of mesenchymal morphology was assessed by three independent and blinded assessors, and interobserver agreement was found to be high (Cohen's kappa $p < 0.001$; Fig. 1c). This scoring revealed a large heterogeneity in the rates at which mesenchymal morphologies appeared. For instance, 031 M cells became mesenchymal after only 3d of CRT, whereas 007B cells only showed marginal changes after 14d of CRT. Of note, the fast EMT onset 031 M cell line also lost mesenchymal morphology after CRT was halted, whereas the slow onset 007B line retained a mesenchymal morphology after treatment cessation (Supplemental Fig. S1a). To support the morphological observations with well-established markers, FACS-based mesenchymal and epithelial markers were assessed in 031M and 007B cells during CRT (Fig. 1d). This confirmed a pronounced induction of vimentin (VIM, a mesenchymal marker) over time in the 031 M cells. This was supported by immunofluorescence, showing a striking loss of cell-cell adhesion marker ZO1 and the appearance of a fibroblast-like cytoskeleton by actin 14d after CRT (Fig. 1e).

Next, cell state markers were measured by FACS before and after CRT in the complete cell line panel. For this, we used mesenchymal

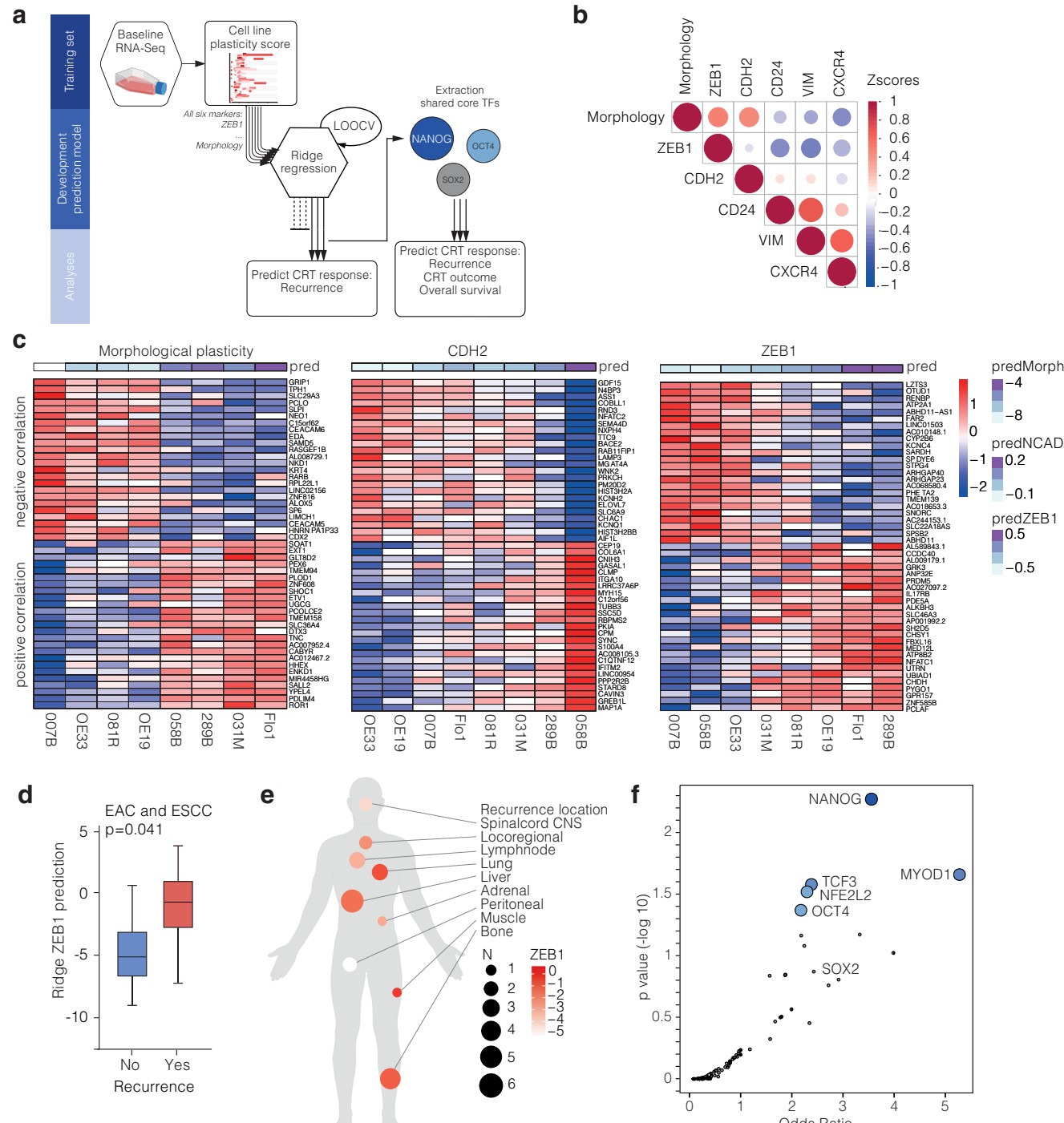

**Fig. 2 | Ridge regression analysis renders predictive models and identifies NANOG and OCT4. a** Flow diagram depicting Ridge regression model development by training of six models. Validation rendered three models with the highest predictive value, subsequently used on pre-treatment biopsies from patients. **b** Ridge regression model correlation matrix plot of all tested in vitro markers. Indicated are *z*-scores of a positive correlation (red) and negative correlation (blue) between markers. Size of dot indicates significance. **c** Heatmaps of top 25 positive and top 25 negative correlated genes with three indicated in vitro markers for all Esophageal adenocarcinoma (EAC) cell lines. Pred = prediction with Ridge regression model. **d** Ridge predicted ZEB1 score with occurrence of recurrences after CROSS-only regimen. *N* = 44; included are EAC and Esophageal Squamous Cell Carcinoma

(ESCC) patients that received CROSS-only neoadjuvant treatment. For EAC patients that received CROSS-only neoadjuvant treatment a similar trend was seen which was not significant; *n* = 33 *p* = 0.09, one-sided Welch t-test, which we attribute to lack of statistical power. **e** Location of distant metastasis after neoadjuvant chemoradiation and resection in EAC patients. Size of dots indicates the number of patients with a metastasis in that location, color indicates ZEB1 prediction score. **f** Volcano plot with transcription factors related to the top 25 positive correlated genes of three in vitro markers as shown in panel E. Inferred with Enrichr enrichment analysis based on 'ENCODE and ChEA Consensus Transcription Factors (TF)' library. Blue dots = significant p-value.

**Fig. 3 | *NANOG* and *OCT4* expression in pre-treatment biopsies are highly predictive for outcome in EAC. a** Box-dot plot of *NANOG, OCT4 and SOX2* expression for pre-treatment Esophageal adenocarcinoma (EAC) biopsies concurrent with available chemoradiation response assessed in resection specimen after chemoradiation therapy (CRT) by Mandard score. Samples sizes indicated and individual data points are shown. Mandard 1 (no tumor left after CROSS), Mandard 2 (major response), Mandard 3 (medium response) and Mandard 4 (minor response). Box indicates mean and interquartile range (IQR), whiskers 1.5x IQR. Lines between two categorized groups indicated if significant, Mann–Whitney *U* tests, *$p < 0.05$, **$p < 0.01$, ****$p < 0.0001$. **b** Box-dot plot of *NANOG, OCT4* and *SOX2* expression in pre-treatment EAC biopsies of patients with or without recurrence after CRT treatment and resections. Samples sizes indicated. Box indicates mean and IQR, whiskers 1.5x IQR. Mann-Whitney U statistical test. **c** Kaplan–Meier survival analysis in all pre-treatment EAC biopsies with known patient follow-up data. Biopsies were dichotomized by median expression of *NANOG, OCT4* and *SOX2*. Survival analysis was performed using Kaplan–Meier analysis and Log-rank statistical test. Patients per group indicated.

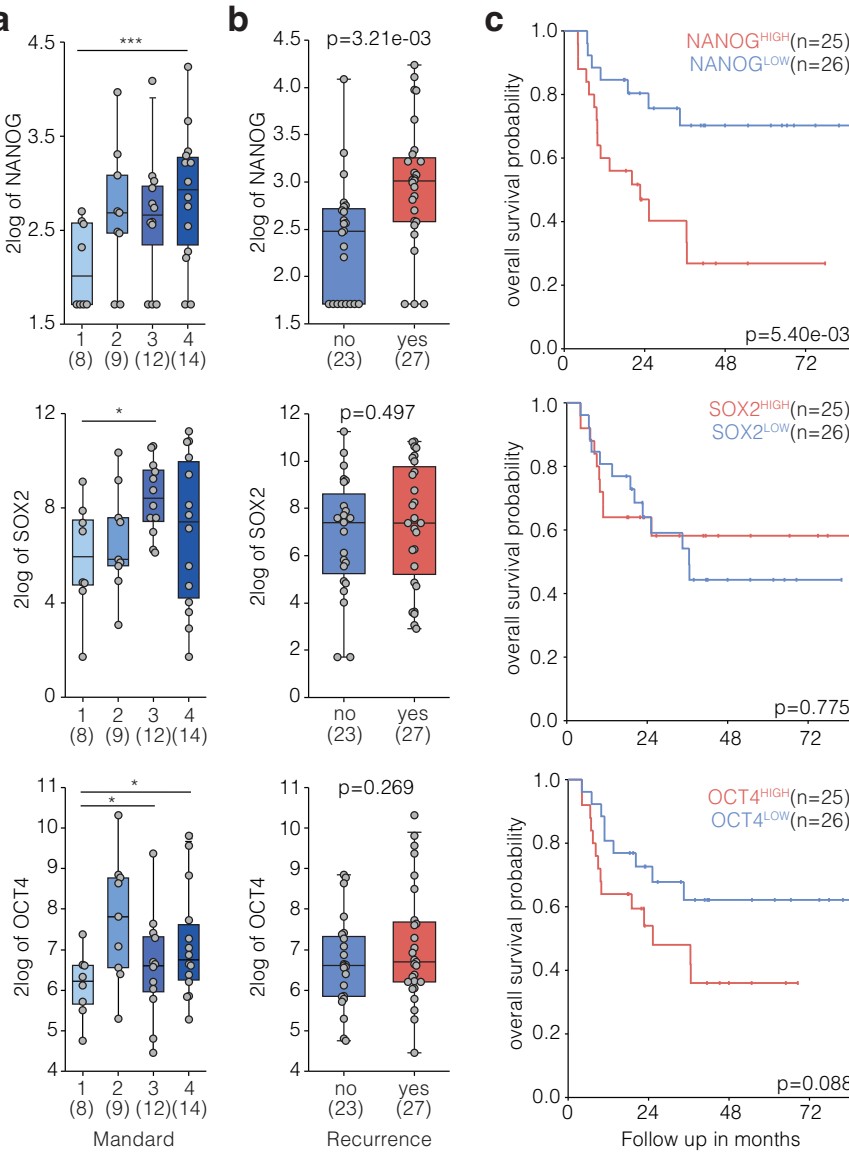

markers zinc finger E-box-binding homeobox 1 (ZEB1), Integrin beta-1 (ITGB1, also known as CD29), N-cadherin (CDH2), C-X-C chemokine receptor type 4 (CXCR4), and epithelial markers E-cadherin (CDH1), cluster of differentiation 24 (CD24), and epithelial cell adhesion molecule (EPCAM). Again, a prominent contrast in the induction of markers was observed between cell lines, which strongly aligned with their ranking based on morphology (Fig. 1f). For instance, 031M cells had the most rapid onset of a mesenchymal morphology, but also the most robust induction of CD29 (and reduction of CDH1) in response to CRT. Conversely, the 007B line was slowest in the morphology ranking and did not show marked shifts in marker expression. Transcript analysis of mesenchymal markers revealed a similar heterogenous response in marker expression following CRT (confirming the morphology- and FACS-based measurements), and shows the robustness of the induction of EMT (Supplemental Fig. S1b). Of note, ESC and EAC subtypes did not differ in the onset of mesenchymal morphology. Neither did prior exposure to chemoradiation or derivation from metastatic lesions (Supplemental Table S2). In addition, the ranking of cell lines by cell viability following CRT did not align with the mesenchymal transition ranking, suggesting that the observed mesenchymal states do not result from selection by CRT and are rather the result of direct induction (Supplemental Fig. S1c, d).

Together, these results show that a substantial heterogeneity in mesenchymal plasticity exists between cell lines, in line with the large differences in outcome observed between EAC patients. We next aimed to leverage this heterogeneity to identify the gene expression programs associated with high propensity for cell state transitions (i.e., high plasticity).

### Ridge regression models for plasticity predict metastatic recurrence in patients

To reveal the gene expression programs that predict the propensity with which mesenchymal transitions occur in vitro, the complete baseline cell line panel was subjected to RNA-sequencing. Principal Component Analysis (PCA) showed that biological replicates and histological subtypes clustered together (Supplemental Fig. S2a). Plasticity rankings did not group together. To formally exclude that baseline mesenchymal, epithelial, or hybrid E/M states predicts plasticity, we first clustered the 8 EAC cell lines in 3 groups (Supplemental Fig. S2b, c). This revealed a cluster of decidedly mesenchymal cell lines at baseline (Flo-1, 298B), highly epithelial cells (007B, 081 R, OE19, 031 M), and an intermediate group (058B, OE33) that we consider hybrid. Expression of canonical E/M markers corroborated this; the intermediate group is characterized by high *ZEB1* gene expression and *EPCAM* gene expression. The mesenchymal cluster is solely high in *ZEB1* and the epithelial group solely high in *EPCAM*. Of note, these clusters did

not align with the plasticity ranking, confirming that baseline mesenchymal or hybrid cell states do not predict propensity for EMT in EAC.

Next, gene expression was correlated to the observed in vitro plasticity by Ridge regression analysis. To generate a prediction model from high-dimensional gene expression data, in which the number of unknown parameters is larger than the sample size, overfitting of the model should be prevented[18,19]. We therefore opted for Ridge regression as this prevents overfitting by estimating a regression coefficient for each predictor variable, rather than discarding them as is done in for instance Lasso regression. Expression data of 8 EAC lines was used as a training set for Ridge regression leave-one-out cross validation (LOOCV) to correlate gene expression patterns to in vitro plasticity marker ranking (Fig. 2a). Six models were trained for the degree to which a marker was induced in response to CRT, or the morphological-based ranking (details in the Methods section). Three highly predictive models were identified based on the assessed in vitro markers; morphological assessment, CDH2, and ZEB1. These models had the highest predictive value and correlated best with each other's predictions (Fig. 2b, and Supplemental Fig. S2d). The top genes indeed reveal good correspondence with the learned phenotype. Models generated from markers CD24, CXCR4 and VIM failed to be consistently predictive. Of the highly predictive models the top 25 positive and negative genes were identified (Fig. 2c).

We next determined the ability of the in vitro-generated plasticity models to predict mesenchymal transitions in esophageal cancer patients. After screening of 139 pre-treatment esophageal patient biopsies, 100 pre-treatment patient biopsies were selected for RNA-sequencing (Supplemental Fig. S3a, b). This stands as the second largest expression dataset of pre-treatment esophageal tissue with treatment response data to our knowledge[20,21]. Baseline characteristics of all patients are listed in Supplemental Table S1 (Supplementary Data 1). Plasticity predictions of the three best performing in vitro-generated models were projected on 44 EAC and ESCC biopsies from patients that later received CROSS-only followed by resection. ZEB1-derived plasticity predictions on pre-treatment biopsy material were best able to predict metastatic recurrences after CRT and surgery in these patients (Fig. 2d). Of note, the most commonly reported metastatic site for esophageal cancer is the liver, followed by lung, bone and brain[22]. We observed the largest number ($N = 6$) of metastases to the liver, in agreement with literature. However, we observed the highest ZEB1 prediction score for metastases to the lung and bone (Fig. 2e). We explain this by the highest ZEB1 prediction score representing the highest mesenchymal plasticity, and therefore the ability to seed to, and survive in, less likely metastatic niches. Of note, expression of ZEB1 per se was not predictive for recurrences, underscoring the additive value of the Ridge regression model.

To consolidate the three separate predictive models (morphological assessment, CDH2, and ZEB1), and to determine whether a shared regulatory mechanism could be identified, we pooled the top-25 positively correlating genes of the three models and performed a transcription factor (TFs) motif enrichment analysis using Enrichr[23,24]. In the top associated TFs, we noticed that pluripotency factors NANOG, OCT4 and SOX2 were represented (Fig. 2f)[25]. We take the overrepresentation of these factors to suggest that pluripotency may be a requirement for mesenchymal transitions.

## NANOG and OCT4 expression is highly predictive for treatment-related outcomes
Several studies have shown that pluripotent cancer stem cell populations arise in cancer following therapy and that these contribute to therapy resistance[26,27]. Given that we found the pluripotency factors NANOG, SOX2, and OCT4 as potential regulators of mesenchymal plasticity, we next aimed to investigate whether their expression predicts clinical variables that relate to therapy resistance and increased tumor cell mobility in EAC. NANOG, SOX2, and OCT4 expression were assessed in pre-treatment biopsies of EAC patients that were subsequently treated with CROSS regimen CRT. For CROSS-treated patients, a pathological response score for chemoradiation was available (Mandard tumor regression scores[28]).

NANOG expression in pre-treatment biopsies was highly predictive of the response to neoadjuvant chemoradiation, and a similar trend was observed for SOX2 and OCT4 (Fig. 3a).

In addition, high pre-treatment NANOG expression associated with recurrence as distant metastases (Fig. 3b) and predicted overall survival (Fig. 3c). Again, SOX2 and OCT4 expression showed similar trends. Another significantly associated transcription factor unrelated to pluripotency, NRF2, was found not to be predictive for treatment outcome (Supplemental Fig. S4). Together, these data show that the pluripotency factor NANOG identified as potential regulators of mesenchymal plasticity in EAC, indeed associates with clinical variables related to both mesenchymal transitions (metastases) and stemness (primary resistance). Although the possibility to detect pluripotency markers on FFPE material would aid clinical implementation, we were unable to achieve convincing results from immunohistochemistry for NANOG, SOX2, or OCT4 (not shown).

## Perturbing pluripotency prevents the onset of mesenchymal states and sensitizes preclinical EAC models to chemoradiation
We next aimed to experimentally determine whether the identified pluripotency factors contribute to plasticity in EAC. While specific pharmacological inhibition of NANOG remains challenging, the inhibitor Niclosamide has been proposed[29]. This FDA-approved and clinically used inhibitor has shown promising anti-tumor activity in several cancers[15]. Cell viability in response to Niclosamide was established per cell line (Supplemental Fig. S5a). Subsequently, the EAC panel was subjected to chemoradiation and the $IC_{20}$ of Niclosamide, which indeed reduced CRT-induced NANOG protein levels and sensitized most cell lines to chemoradiation (Supplemental Fig. S5b, c). In agreement, an inhibition of CRT-induction mesenchymal markers by Niclosamide was observed (Supplemental Fig. S5d). This was confirmed by immunofluorescence for VIM in 289B cells (Supplemental Fig. S5e).

To further ascertain whether these inhibitors could be sensitizers for (chemo)radiation in preclinical models for EAC, we turned to a previously described EAC organoid lines (Pt382[12]) and a new patient-derived organoid line Pt376. Organoids were plated and treated with 1 Gy radiation daily and 300 nM Niclosamide. Cell viability was assessed after a week of treatment (Supplemental Fig. S6a, b). This revealed that the compound strongly sensitized EAC organoids to radiation. Next, we aimed to assess if Niclosamide could radiosensitize tumor cells in vivo. We previously observed that perturbation of resistance mechanisms dramatically improved responses to radiation therapy particularly in the 081R cell line[30]. 081 R cells were subcutaneously grafted in the hind limb of NSG mice to allow localized radiation therapy as previously published by us. However, compared to irradiated mice that received vehicle, Niclosamide did not result in a significant inhibition of tumor growth and as a consequence, maximal tumor sizes were reached at approximately similar times (Supplemental Fig. S6c, d).

Given the inconclusive in vivo results using pharmacological inhibition, we set up to investigate the consequences of silencing NANOG and OCT4. To do so, NANOG and OCT4 genes were targeted by lentiviral shRNA delivery in the 081 R line, and knockdown was validated (Supplemental Fig. S7a). Clones TRCN0000004880 (shNANOG) and TRCN0000004886 (shOCT4) were selected based on knockdown efficiency for further experiments. TGFβ is a well-known inducer of mesenchymal cell states. This induction was effectively prevented in NANOG and OCT4 silenced cells treated with TGFβ (Fig. 4a, b). We confirmed that shNANOG and shOCT4 knockdown prevent cell state transitions, as mesenchymal markers showed minor increases in response to CRT (Fig. 4c and Supplemental Fig. S7b). To confirm that the link between mesenchymal plasticity and the identified TFs is specific to pluripotency factors such as NANOG and OCT4, we also silenced NFE2L2 (NRF2) (Supplemental Fig. S8a). Here, we did not observe marked differences in EMT induction between scrambled control and shNFE2L2 (Supplemental Fig. S8b). However, we also measured cell confluency during chemoradiation and observed that NRF2 protects against this treatment (Supplemental Fig. S8c). We explain this by

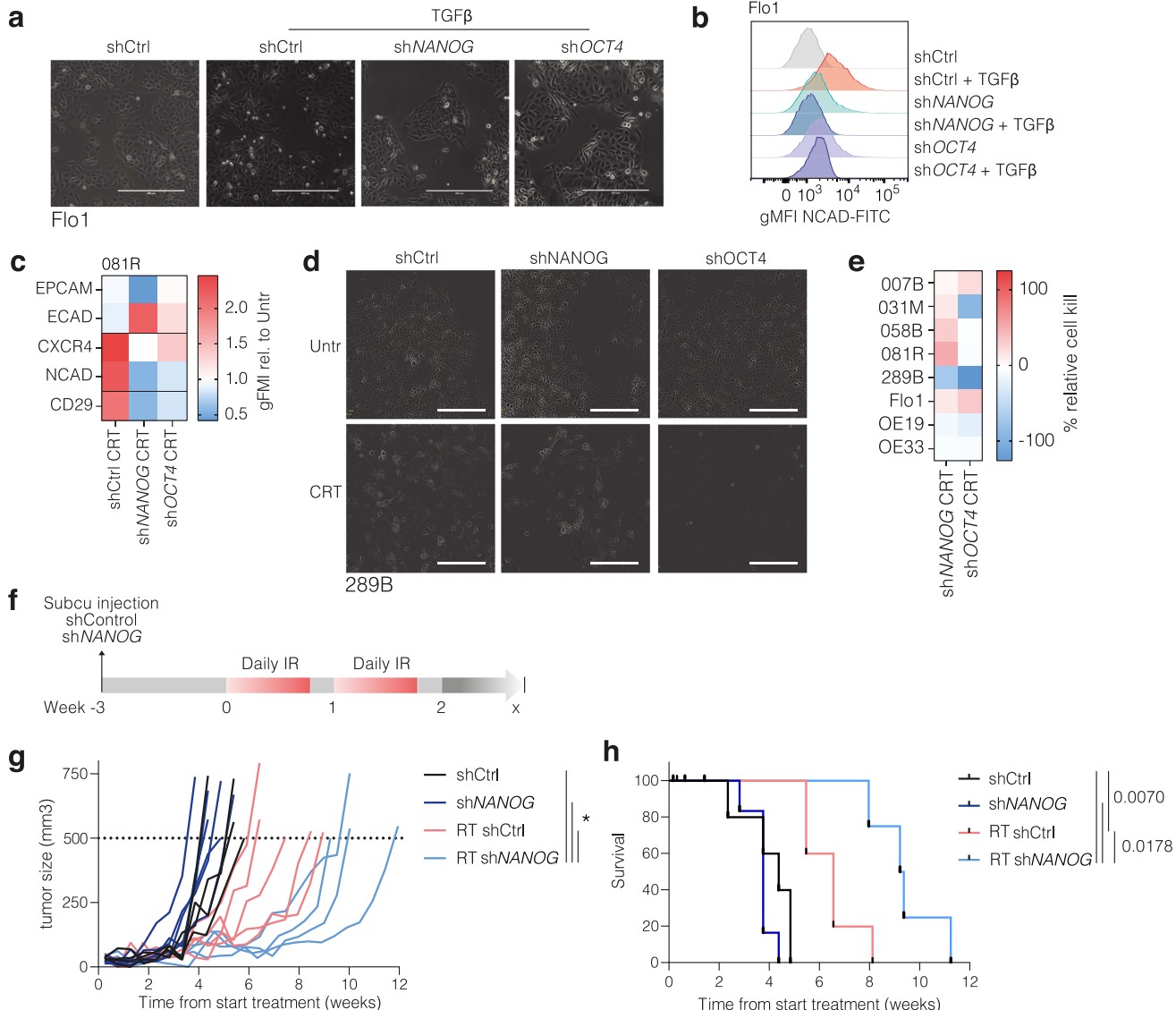

**Fig. 4 | Inhibition of NANOG and OCT4 by KD reduces plasticity and sensitizes EAC models for chemoradiation. (a)** Phase-contrast images showing induction of mesenchymal morphology in Flo1 cells silenced for pluripotency factors (*shNANOG* and *shOCT4*), compared to scrambled control (shCtrl) after 5 day exposure to 5 ng/mL TGFβ. **(b)** Flow cytometry analysis of expression of the mesenchymal marker NCAD of cells shown in panel A. **(c)** Heatmap of relative geometric Mean Fluorescent Intensity (gMFI) of epithelial (EPCAM, ECAD) and mesenchymal (CXCR4, NCAD, CD29) markers in 081 R cells treated with chemoradiation (CRT). Average of biological duplicates of knockdown *NANOG* and *OCT4* compared to scrambled control cells. **(d)** Example of 289B cells as in panel C, phase-contrast images of silenced cells before (Untreated) and after 14 days treatment of CRT. **(e)** Relative confluence compared to control of Esophageal adenocarcinoma (EAC) cells treated with CRT for 14 days, knockdown of *NANOG* and *OCT4* compared to scrambled control cells. Average of biological duplicates. Statistical test used is Mann Whitney U. **(f)** Schematic of setup for radiation treatment of NOD scid gamma mouse (NSG)

mice, and treatment schedule showing grafting, treatment, and follow-up. Tumor is grafted on hind limb (1×10⁵ cells in 50% Matrigel). At a predetermined start time corresponding to tumor sizes of approximately 100 mm3, treatments commenced. Radiation was 2×5 consecutive days, 2 Gy per fraction to a cumulative dose of 40 Gy. Tumor volumes were measured continuously. Once tumors reached 500mm³, the humane endpoint was reached and mice were culled. **(g)** Tumor volumes over time from start of injection for each treatment group. Lines indicate individual mice. Group size N = 6, mice that reached a humane endpoint other than tumor volume 500mm³ as determined a priori were excluded, resulting in N = 4-5 per group for analysis. Lines between groups indicated when significant, * p < 0.05. **(h)** Kaplan-Meier survival analysis of mice over time. Events are humane endpoints by maximum tumor growth. Group size N = 6, censored mice indicated in black short lines, resulting in N4-5 per group for analysis. Lines indicated if significant, p-values by log-rank test indicated for groups of interest.

the fact that radiation induces ROS, and that NRF2 reacts to ROS by inducing mitochondrial biogenesis. The latter has recently been shown by us to be an additional resistance mechanism in EAC[30].

Next, *NANOG* and *OCT4* were silenced across the full cell line panel, and cell viability following CRT was determined. This revealed that *NANOG* silencing most consistently sensitized cells to CRT (Fig. 4d, e). To establish whether silencing NANOG also sensitized to therapy in vivo, 081R *shCtrl* and *shNANOG* cells were grafted in the hind limb of NSG mice (as for

Supplemental Fig. S6). Tumors were treated with radiation for 14d and mice were followed up (Fig. 4f). Compared to irradiated *shCtrl* tumors, irradiated *shNANOG* tumors were strongly delayed in their growth, and maximal tumor sizes were reached much later (Fig. 4g, h). Of note, *shNANOG* did not delay the onset and rate of tumor growth in the absence of radiation, suggesting its role in EAC to be limited to responses to therapeutic stress. Together, these data show that pluripotency factors are important for chemoradiation-induced cell state transition in EAC. Their perturbation

prevents mesenchymal states from occurring and sensitizes EAC tumors to radiation. In anticipation of effective pharmacological interventions, the use as predictive biomarker for EAC patients could be envisioned.

## Discussion

Incomplete responses to neoadjuvant chemoradiation in esophageal cancer may lead to tumor recurrence and metastatic disease, even despite the recently observed benefit of adjuvant immunotherapy[3,31]. Therapeutic pressure has been established as a contributor to metastatic disease[10,32]. Especially in EAC, cells appear to harbor a high degree of plasticity and quickly adopt a mesenchymal cell state[10,16,33]. Thus far, the regulators of this plasticity in EAC were unknown. In this study, we identified pluripotency factors NANOG and OCT4 as drivers of high plasticity in EAC cell lines and found them to be predictive for outcome after chemoradiation in EAC patients: Mandard score and overall survival.

Cancer stem cells have been reported to drive esophageal cancer growth and resistance[34]. Markers such as ALDH1, NANOG[35], OCT3/4[36], and SOX2[30,31] have been associated with cancer recurrence and therapy resistance. In several other cancer types, NANOG overexpression was correlated with increased metastatic potential and proliferation in cancer[37–39]. Of note, a vast body of literature describes the occurrence of stemness markers and properties as a consequence of mesenchymal transitions rather than a prerequisite. Instead, we observed that classic pluripotency factors are required for mesenchymal transitions, and that their inhibition prevents mesenchymal states in EAC. This is in line with a limited number of studies in ovarian cancer, hepatocellular carcinoma and bladder cancer, where the ability to undergo mesenchymal transitions was shown to require NANOG[40–42].

Two scenarios could explain the association of pluripotency factors with mesenchymal states: One is that chemoradiation selects for pre-existing therapy resistant stem-like cells[39]. These likely also have mesenchymal features (as described) and their enrichment by therapeutic pressure will result in a population that is more mesenchymal as a whole. However, the observed rapid and wide-spread appearance of mesenchymal cells argues against selection and favors a model in which mesenchymal states are instructed. We, therefore, hypothesize a second scenario in which pluripotency factors are required to instill a permissive cell state that allows such transitions to take place. It would be interesting to interrogate the epigenetic landscapes that associate with these permissive states[43,44]. Mesenchymal transition is often considered a switch-like transition. However, there is a growing body of evidence for the existence of a gradient of intermediate E/M states. These hybrid states express both epithelial and mesenchymal markers, and behave more like mesenchymal cells while retaining their epithelial identity[12]. Recent studies show that hybrid E/M states are more efficient in forming metastases, compared to the extremes in this transition, i.e., fully epithelial or mesenchymal[45]. Also, recent work has indicated that fully mesenchymal cells are in fact less plastic than hybrid cells that derive from epithelial cells[46]. This is in line with our data, as the propensity for plasticity rather than a mesenchymal baseline state associates with worse prognosis.

In the present study we found that pharmacological inhibition of pluripotency factors prevented transitions to a mesenchymal phenotype, and sensitized cells to chemoradiation in vitro. In the clinic, targeting NANOG is promising as its expression is relatively limited to for instance embryonic development, and cancer cells[37,38]. This suggests that efficacy could be achieved with low toxicity. The pluripotency inhibitor Niclosamide have shown promising preclinical anti-tumor activity in several cancers, including activity against EMT[29,47–51]. However, Niclosamide was not significantly effective at sensitizing patient-derived grafts to radiation in our hands. The reasons for this are unknown but possibly its pharmacokinetics are unfavorable. In addition, specificity is questionable, as the inhibitor has additional mechanisms of action unrelated to pluripotency[52]. Possibly, by interrogating the permissive cell state that is required for mesenchymal transitions, targetable molecules that act at other biological levels may become apparent in future research. Epigenetic readers or non-coding genomic and transcriptomic elements for which RNA-based interventions are available could further elucidate this interplay. Also, we have to acknowledge that EMT is not the sole contributor to therapy resistance, and the other processes such as metabolic rewiring are at play[30]. Having stated that, these can be argued to also require plasticity and may also rely on pluripotency factors.

Despite not providing a promising pharmacological intervention, our study does provide important novel insights in the mechanisms that explain the high plasticity of EAC cells. Additionally, NANOG expression in pre-treatment EAC biopsies was highly predictive for response to therapy and therefore patient outcome. We propose that this knowledge, together with the development of predictive biomarkers for patient selection, or accurate treatment monitoring tools could be used to improve the efficacy of chemoradiation in EAC.

## Data availability

Numerical results underlying graphs in this manuscript is available in Supplementary Data 2. Gene expression data are accessible at GEO under GSE254942, and on Figshare: https://figshare.com/s/2cf7cd28805715224594. All reagents and data can be requested from the corresponding author upon reasonable request.

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

## Acknowledgements

We thank the patients for participating. This work was supported by Oncode funding to J.P.M. and a KWF Dutch Cancer Society project grant (10992 / 2017-1) to M.F.B. and H.W.M.L.

## Author contributions

Design research: A.P.Z., M.F.B., and H.W.M.L. Data interpretation, analysis, generation, and/or collection: A.P.Z., M.P.G.D., P.M., H.B., R.J., P.B., J.K., D.Z., R.V., C.W., M.v.M., D.B., T.v.d.E., C.O., S.D., A.C., E.A.E., G.K.H., S.L.M., M.I.v.B.H., J.P.M., H.W.M.L., and M.F.B. Supervised the project: J.P.M., H.W.M.L., and M.F.B. Wrote the paper: A.P.Z., M.F.B., and H.W.M.L.

## Competing interests

M.F.B. has received research funding from Celgene, Frame Therapeutics, and Lead Pharma, and has acted as a consultant to Servier and Olympus. H.W.M.L. Consultant or advisory role: BMS, Daiichy, Dragonfly, Eli Lilly, MSD, Nordic Pharma, Servier. Research funding and/or medication supply: Bayer, BMS, Celgene, Janssen, Incyte, Eli Lilly, MSD, Nordic Pharma, Philips, Roche, Servier. Speaker role: Astellas, Daiichy, Novartis. J.P.M. has acted as a consultant to AbbVie. M.I.v.B.H. served as consultant for Johnson and Johnson, Medtronic, Alesi Surgical, Mylan and has received unrestricted research funding from Stryker. None of these parties were involved in the design of this study or drafting of the manuscript. All other authors have no competing interests to declare.
