## [Peer Review File · Communications Medicine]

Reviewers' comments:

Reviewer #1 (Remarks to the Author):

The authors map the association between stemness marker NANOG and mesenchymal induction, in response to chemoradiotherapy (CRT) through tracking cell-state transitions in multiple esophageal cultures. They identified specific transcription factors including NANOG which played a role in enabling these cell-state transitions and also correlate with clinical outcomes. The study is overall well-designed, and addresses an important theme of connecting EMT and stemness with therapy resistance. I have the following suggestions for the authors to consider:

1. The study highlights a range of plasticity seen in the multiple cultures in response to CRT. The authors should consider discussing their observations in the context of these recent studies on 'resistance to EMT' or 'resistance to MET' ('irreversible EMT'). Also, can the authors investigate whether the proclivity to undergo EMT depends on their pre-treatment phenotype along the epithelial-hybrid-mesenchymal cell-state spectrum, given that hybrid E/M cells have been shown to be more plastic than full EMT or full MET ones?

<https://pubmed.ncbi.nlm.nih.gov/36628532/>

<https://www.science.org/doi/10.1126/sciadv.abj8002>

<https://elifesciences.org/articles/76535>

<https://pubmed.ncbi.nlm.nih.gov/29670281/>

<https://www.biorxiv.org/content/10.1101/2020.03.19.998823v2>

2. The authors also report NFE2L2 (NRF2) as a transcription factor, but do not investigate it further. NRF2 has been shown to stabilize hybrid E/M states, which are often more stem-like or metastatic than the terminal epithelial or mesenchymal ones. Thus, the authors should investigate the impact of NFE2L2 in preventing cell-state transitions, and assess whether the levels of NFE2L2 correlate with survival in the tumour biopsies they have.

Relevant papers related to NRF2 and higher stemness in hybrid E/M cells are given below:

<https://pubmed.ncbi.nlm.nih.gov/31329868/>

<https://www.pnas.org/doi/10.1073/pnas.1812876116>

<https://cancer.cbiomedcentral.com/articles/10.1186/s12935-021-01822-1>

<https://pubmed.ncbi.nlm.nih.gov/34847378/>

Reviewer #2 (Remarks to the Author):

The authors hypothesize that the induction of EMT including occurrence of stem cell like cell propensities is responsible for resistance to neoadjuvant (radio) chemotherapy in esophageal adenocarcinoma

Using cell line, tissue based, and animal models they identified NANOG as a key player for prediction of EMT and survival.

The setup of the study is very focused on mesenchymal transition/EMT which is fine. However, the authors should also discuss other potential mechanisms for response (tumor – host interaction, TME, senescence, etc). It should also be discussed whether their approach, including EMT as given

factor for resistance does not imply a bias towards the impact of EMT in the context of therapy resistance.

The study is well conducted and technically firm. The results are interesting and would open new insights of the highly relevant clinical problem of therapy resistance to conventional treatment in EAC.

I have, however some remarks:

1) The major shortcoming of this study, however, that further validation of the observed in vitro / animal models is somehow lacking: I would strongly recommend to confirm the results in patient material, i.e. analyzing post-treatment tumor tissue samples and comparing pre- and post therapeutic status; primary resected (historic) vs. treated; different samples with different responses including major (complete would not be available) and minor or no regression. I assume that additional FFPE material is available.

2) Moreover, some ESC cell lines were also included which is only addressed to very briefly. Are the results of these analyses also included in the results that later form the base of the final conclusions? Please provide a more detailed description of the primary EAC cell lines used (tumor type, treatment) as well and discuss potential influences on the results

3) Tissue Analysis: the presentation of the case numbers is confusing. As far as I can see, the conclusion that NANOG expression is associated with tumor regression bases on the analyses of tissue of 44 patients (or 34 – according to figure 3), who were treated with CROSS CRT but not of all 78. This is misleading in the abstract and elsewhere in the text. Moreover, supplemental table 1 indicates that not all patients received CRT. This should be clarified. Also, in table 1 which is important for a clean dataset, there are some other issues to be clarified: e.g., one case is labelled as stomach cancer, which would be an exclusion criterium? T and N-stage: clinical or pathological/post-therapeutic? Please indicate this appropriately – comparing the data about Mandard grades it is probably pre-therapeutic clinical staging? Data about regression should also be included in table 1. Were there no tumors with Mandard 5? In the section regarding patient biopsy processing, 139 biopsies and 105 resection specimens were listed. Supplemental figure 3 lists 168 cases. I do not see any resection specimens used for analyses, or were the primary cultures from this group? This part should be presented clearer.

4) How would the RNA/RNAseq data transfer to protein expression – e.g. immunohistochemistry? For a promising biomarker, an easy application in clinical routine would make sense.

Reviewer #3 (Remarks to the Author):

The manuscript describes developing a model that predicts the response of EACs to CRT. This reveals NANOG and SOX4 as key associated transcription factors. Perturbation of NANOG was shown to reduce post-CRT viability, indicating that NANOG could be a biomarker for outcome prediction. The study is clearly written and well-performed with elegant experiments. Sometimes the moving between different cell lines makes the story somewhat fussy as this is not always well explained. Another point that needs to be discussed is that the data do not reveal whether the recurrences are CRT induced or may have been already present (at very low levels) before the treatment. This should at least be discussed.

Figure S1a, due to the few cells shown, it is unclear that 031M lost mesenchymal morphology, whereas 007B retained it. It looks like 031M already had a more MSC appearance without IR. Quantify and explain better.

Fig1D What were the absolute differences between the cell lines? I suppose 031M already had a lot higher level of Vim?

Fig1. d, mention below fig 1e should be Fig 1f? otherwise, the text above does not make sense.

Fig S1b. The markers are very inconsistent between cell lines. This should be mentioned.

Fig2d label the x-axis as metastasis recurrences.

Fig2e, this is unclear. How can be distinguished between CRT-induced and already present metastasis? Why in the lung and not in other places? Is it related to ZEB1?

Fig S4e, why suddenly 289B cells here? They were already low in confluency. The drug seems to have little effect on them. How about the other cell lines?

FigS4f cand related sentence can be removed as it does not mean anything.

With FigS5a,b - both compounds? I see only NIC.

Top line 1 of page 12 Fig 4d, eE? Capital E can be removed?

REVIEWER 1

The authors map the association between stemness marker NANOG and mesenchymal induction, in response to chemoradiotherapy (CRT) through tracking cell-state transitions in multiple esophageal cultures. They identified specific transcription factors including NANOG which played a role in enabling these cell-state transitions and also correlate with clinical outcomes. The study is overall well-designed, and addresses an important theme of connecting EMT and stemness with therapy resistance. I have the following suggestions for the authors to consider:

1.1. *The study highlights a range of plasticity seen in the multiple cultures in response to CRT. The authors should consider discussing their observations in the context of these recent studies on 'resistance to EMT' or 'resistance to MET' ('irreversible EMT'). Also, can the authors investigate whether the proclivity to undergo EMT depends on their pre-treatment phenotype along the epithelial-hybrid-mesenchymal cell-state spectrum, given that hybrid E/M cells have been shown to be more plastic than full EMT or full MET ones?*

We thank the reviewer for this suggestion. We had not thought of it initially and agree that it is relevant to the paper. The following is now added to the discussion:

“Mesenchymal transition is often considered a switch-like transition. However, there is a growing body of evidence for the existence of a gradient of intermediate E/M states. These hybrid states express both epithelial and mesenchymal markers, and behave more like mesenchymal cells while retaining their epithelial identity (1). Recent studies show that hybrid E/M states are more efficient in forming metastases, compared to the extremes in this transition, i.e., fully epithelial or mesenchymal (2). Also, recent work has indicated that fully mesenchymal cells are in fact less plastic than hybrid cells that derive from epithelial cells (3). This is in line with our data, as the propensity for plasticity rather than a mesenchymal baseline state associates with worse prognosis.”

Of note, we did not find associations between pre-treatment mesenchymal (or hybrid) state and their propensity to undergo EMT. We now labelled the PCA plot, originally found in Figure S2A, with the plasticity rankings (**Figure R1a**, bold numbers indicate ranking). We have added a panel of images from all 8 EAC cell lines to allow interpretation of the baseline phenotype versus the degree of plasticity (**Figure R1b**).

Figure R1 | Baseline characteristics of the EAC cell line panel

(a) Principal component analysis (PCA) plot showing cell line panel samples in biological replicates. Bold numbers indicate plasticity ranking of each cell line (1, fastest; 12 slowest).

(b) Phase-contrast images of baseline phenotype of the panel of eight EAC cell lines, sorted by visually scored epithelial to mesenchymal state. Scale bar is 200 μm . Bold numbers indicate plasticity ranking.

To formally ascertain whether hybrid E/M states associate more with plasticity than completely epithelial or mesenchymal states do, we first decided to cluster the cell lines in 3 groups (**Figure R2**). This revealed a cluster of decidedly mesenchymal cell lines at baseline (Flo-1, 298B), highly epithelial cells (007B, 081R, OE19, 031M), and an intermediate group (058B, OE33) that we consider “hybrid”. Expression of canonical E/M markers corroborated this; the intermediate group is characterized by high *ZEB1* gene expression and *EPCAM* gene expression. The mesenchymal is solely high in *ZEB1* and the epithelial group solely high in *EPCAM*. Also, these clusters align with the visually ranked baseline mesenchymal states (Figure R1b).

Of note, these clusters did not align with the plasticity ranking, again confirming that baseline cell states (mesenchymal or hybrid) do not predict propensity for EMT in EAC.

Figure R2 | Identification of intermediate, hybrid E/M EAC cell lines and their association with plasticity.

The consensus matrix at $k=3$ depicts three distinct cluster groups: epithelial, intermediate, and mesenchymal. Consensus values within this matrix span a spectrum from 0 (indicating that elements are never clustered together) to 1 (signifying that they are always clustered together), represented by a gradient from white to dark blue. These results consistently align with the grouping observed in Figure R1a (PCA). Plotted above are canonical epithelial and mesenchymal markers. Plasticity ranking is shown in top row. Note that darker color indicates slower EMT onset.

Furthermore, cell line derivation site (primary or metastasis), and prior treatment exposure were all independent from the propensity to undergo EMT (**Table R1**). The modifications and additions are also added to the revised figures as **Figure S2a-c**. In addition, we now include a full description of the primary cell lines including prior treatment status, their plasticity ranking and whether they are derived from the primary tumor or metastatic lesion in the revised manuscript (new **Table S2**).

1.2. *The authors also report NFE2L2 (NRF2) as a transcription factor, but do not investigate it further. NRF2 has been shown to stabilize hybrid E/M states, which are often more stem-like or metastatic than the terminal epithelial or mesenchymal ones. Thus, the authors should investigate the impact of NFE2L2 in preventing cell-state transitions, and assess whether the levels of NFE2L2 correlate with survival in the tumour biopsies they have.*

Relevant papers related to NRF2 and higher stemness in hybrid E/M cells are given below:

This is an interesting suggestion that we have now incorporated (**Figure R3** below and Supplemental **Figure S4** in the revised manuscript). *NFE2L2* in pre-treatment biopsies does not strongly associate with clinical variables related to therapy response and outcome. Of note, other candidate TFs from Figure 2f (TCF3 and MYOD1) did not associate significantly with these variables either (not shown).

Figure R3 | Association of NRF2 with outcome variables.

As for Figure 3, using *NFE2L2* (the gene for NRF2). Numbers have been updated in response to Reviewer 2's comments.

In addition, we generated *NFE2L2*/NRF2-silenced cells. From the five shRNA sequences, only one resulted in a downregulation of *NFE2L2* (Figure R4a). We repeated the experiments performed for the *shNANOG* and *shOCT4* experiments in the paper, which was to measure EMT by FACS after two weeks of chemoradiation. Here, we did not observe marked differences in EMT induction between scrambled control and *shNRF2* (Figure R4b). However, we also measured cell confluency during chemoradiation. Here, we did observe that NRF2 protects against chemoradiation (Figure R4c). We explain this by the fact that radiation induces ROS, and that NRF2 reacts to ROS by inducing mitochondrial biogenesis. The latter has recently been shown by us to be an additional resistance mechanism in esophageal cancer (4). We have included these data as Figure S8a-c in the revised manuscript.

Figure R4 | Silencing *NFE2L2* does not prevent EMT but does sensitize to chemoradiation

(a) *NFE2L2* gene expression was measured by RT-qPCR after transducing with *NFE2L2*-targeting shRNA or scrambled control (shc002) and puromycin selection. Bar graphs show means of technical triplicates \pm SD. Statistical test is unpaired Student's t-test.

(b) Fold change of epithelial and mesenchymal markers after 2 weeks of chemoradiation with *NFE2L2*-silenced O81R cells and scrambled controls. Bar graphs show means of biological triplicates \pm SD.

(c) Confluency over time relative to $t=12h$ after seeding. Cells in the chemoradiation (CR) group were treated from two weeks: the black arrows indicate the start of the treatment with carboplatin, paclitaxel and daily radiation, and the gray arrow indicate the start of the drug holiday. Dots show means of biological triplicates \pm SD.

REVIEWER 2

The authors hypothesize that the induction of EMT including occurrence of stem cell like cell propensities is responsible for resistance to neoadjuvant (radio) chemotherapy in esophageal adenocarcinoma. Using cell line, tissue based, and animal models they identified NANOG as a key player for prediction of EMT and survival.

The setup of the study is very focused on mesenchymal transition/EMT which is fine. However, the authors should also discuss other potential mechanisms for response (tumor – host interaction, TME, senescence, etc). It should also be discussed whether their approach, including EMT as given factor for resistance does not imply a bias towards the impact of EMT in the context of therapy resistance.

We thank the reviewer for this point and totally agree that mesenchymal transition/EMT certainly is not the only factor contributing to therapy resistance. In fact, we have previously found that increased oxidative phosphorylation contributes to resistance to radiation in EAC (4). However, in several previous studies we consistently observed that EMT is a recurring and likely predominant resistance mechanism.

For instance, we demonstrated that the tumor microenvironment (TME) contributes to chemoradiation resistance in EAC by inducing EMT (5). Also, we have shown that this transition is initiated in response to chemoradiation and long-term RTK inhibition (6,7). Given these considerations, we considered EMT the prime resistance mechanism to study for the current manuscript. We mention these considerations in the revised Introduction.

The study is well conducted and technically firm. The results are interesting and would open new insights of the highly relevant clinical problem of therapy resistance to conventional treatment in EAC.

I have, however some remarks:

2.1. The major shortcoming of this study, however, that further validation of the observed in vitro / animal models is somehow lacking: I would strongly recommend to confirm the results in patient material, i.e. analyzing post-treatment tumor tissue samples and comparing pre- and post therapeutic status; primary resected (historic) vs. treated; different samples with different responses including major (complete would not be available) and minor or no regression. I assume that additional FFPE material is available.

The reviewer raises an important point. Comparing pre- and post-chemoradiation samples in patients would be the optimal setup to test this. We previously attempted to obtain from our large biobanked cohorts matched pre-treatment biopsies and post-chemoradiation resection specimens for RNA-seq, and only managed to obtain six matched samples (4). In this small set, when we applied GSEA using an EMT gene set we found a non-significant enrichment for EMT gene expression in the resection specimens (**Figure R5**). We believe that a lack of statistical power is the cause of this.

Figure R5 | Gene set enrichment analysis on matched pre- and post-nCRT samples

Gene set enrichment analysis was performed on the matched pre- and post-treatment samples from (4). Resection specimens (post-nCRT) are on the left. Normalized enrichment score was 0.9, adjusted P-value is 0.8.

A possible solution to this would be the use of FFPE material from many more pre- and post-treatment samples. Related to point 4 from this reviewer, we did attempt immunohistochemical staining for the predictive biomarkers (NANOG, SOX2 and OCT4). However, due to technical issues we were unable to address this comment satisfactory (**Figure R6**). SOX2 and OCT4 stainings both yielded very low signals. NANOG showed positive signal in the cytosol which is at odds with its function as a transcription factor function in the nucleus.

Figure R6 | Immunohistochemistry for NANOG, SOX2 and OCT4.

FFPE slides from 5 patients were stained for NANOG, SOX2 and OCT4 by immunohistochemistry. Sections were rehydrated according to routine procedures and antigen retrieval was done by heating at 95°C for 20' followed by cooling down to 75°C for 50' in 10mM sodium citrate (pH 6). Slides were blocked using UltraV blocking reagent (ThermoFisher). Antibodies were added anti-Nanog (4893 / 1E6C4 Cell Signaling, 1:200), anti-Sox2 (2683-1 / EPR3131 BioConnect, 1:200), anti-Oct4 (2750S, Cell Signaling, 1:1000) overnight. Dilutions are based on titrating to below concentrations that give background staining. Following washing, secondaries used were Powervision Poly-HRP-GAM/Rabbit IgG or Poly-HRP-Goat anti rabbit IgG. Endogenous peroxidase was blocked. Detection was with DAB, counterstaining was with Myers Hematoxylin. Slides were dehydrated and mounted in Pertex.

In addition, we found that measuring EMT (as an outcome variable) on FFPE is challenging, which is also observed and concluded in other papers (8,9). Mesenchymal tumor cells are hard to distinguish from adjacent stromal cells: the morphology and molecular markers, such as Vimentin (mRNA and protein) show too much overlap. Projecting an EMT gene set on patient samples is always confounded by stroma (10).

Instead, we would like to point out that validation in patients is in fact provided in Figures 2 and 3 where we established a large cohort of RNA-Sequenced pretreatment biopsies with long-term clinical follow up. We feel that the predictive marker NANOG was validated to a sufficient extent in patient material.

2.2. Moreover, some ESC cell lines were also included which is only addressed to very briefly. Are the results of these analyses also included in the results that later form the base of the final conclusions?

We thank the reviewer for this comment. To clarify, the final conclusions are based on EAC. We indeed took along ESC in Figure 1 and Figure 2, because both EAC and ESC receive the same first-line treatment (neoadjuvant chemoradiation) and disease recurrence remains an issue in both cancer types. Yet, we agree that EAC and ESC are two distinct disease entities, and the validation efforts shown in Figure 3 and following are all on EAC. We have modified the figure legends to more clearly indicate this.

Please provide a more detailed description of the primary EAC cell lines used (tumor type, treatment) as well and discuss potential influences on the results

A detailed description of the primary EAC cell lines was published before in (4), and now also shown in **Table R1**, **Table S2**. Only one primary cell line was previously exposed to chemoradiation before cell line establishment. Two lines were derived from metastatic lesions. These factors can impact the characteristics of the cell line because the tumor cells may have gone through selection of treatment resistance or metastatic dissemination. However, we believe this has a minor effect on the obtained results. The reasons are two-fold: First of all, we did not find any indication that prior treatment or metastasis caused bias in the plasticity ranking (**Table R1** and **Table S2**). Secondly, we used the cell line panel to tell us what features are predictive for plasticity. As long there is heterogeneity in the degree of plasticity, the cell line panel is useful. This is now mentioned in the results.

2.3. Tissue Analysis: the presentation of the case numbers is confusing. As far as I can see, the conclusion that NANOG expression is associated with tumor regression bases on the analyses of tissue of 44 patients (or 34 – according to figure 3), who were treated with CROSS CRT but not of all 78. This is misleading in the abstract and elsewhere in the text.

We agree and have made sure the right patients were included and redid the analyses. In Figure 3a, we made sure to include those EAC patients that received a chemoradiation-based therapy followed by resection (this is necessary to determine response to CRT). This includes CROSS-only but also CROSS with add on therapies (with Trastuzumab and Pertuzumab, and with Nivolumab). These total **47**.

For Figure 3b,c we also included patients that received definitive CRT (who are not resected), as clinical outcome variables were available for these. In short; all patients included in the analysis in Figure 3 received some sort of CRT, and those who did not were excluded. These total **55**.

Note that for some samples variables are unavailable. For instance, Mandard scores are missing for **4** patients (leaving 43 patients), recurrence data is missing for **1** (50 left for analysis), and survival data **4** (55 left). These numbers are now explained in much more depth in the Supplemental Table S1, and the revised Methods (Statistics) section.

The number **44** in Figure 2f refers to all CROSS-only treated EAC and ESCC cases together. If the EAC cases only are considered, the analysis returns a non-significant result that we attribute to the low sample size of 33. We mention these results in the revised figure legend.

Moreover, supplemental table 1 indicates that not all patients received CRT. This should be clarified. Also, in table 1 which is important for a clean dataset, there are some other issues to be clarified: e.g., one case is labelled as stomach cancer, which would be an exclusion criterium?

See our comment above. Indeed, some patients were biopsied but did not receive nCRT afterwards for clinical reasons. Gastric cancers close (proximal) to the gastroesophageal junction are treated as esophageal cancer in the Netherlands.

T and N-stage: clinical or pathological/post-therapeutic?

These are at the moment of diagnosis; clinical. Thank you for pointing this out. Table S1 has been modified to better indicate this.

Please indicate this appropriately – comparing the data about Mandard grades it is probably pre-therapeutic clinical staging?

No, these are response grades determined post-resection.

Data about regression should also be included in table 1.

Indeed, this variable was lacking and we have now added it.

Were there no tumors with Mandard 5?

Indeed, there were no EAC cases with Mandard 5.

In the section regarding patient biopsy processing, 139 biopsies and 105 resection specimens were listed. Supplemental figure 3 lists 168 cases.

We apologize for any confusion. Note that *patient* numbers and *biopsy* numbers do not match perfectly. We have changed Supplemental Figure S3 to precisely indicate the number of *biopsies*. In the downstream analyses, individual patients were used. We would also like to stress that the point of Supplemental Figure S3 was to provide an overview of biopsy quality control for RNA-Seq analysis, and not an overview of patient baseline characteristics.

Note that no resection specimens were processed for RNA-Sequencing but were seen by a pathologist to determine Mandard response grade.

I do not see any resection specimens used for analyses, or were the primary cultures from this group? This part should be presented clearer.

Resection specimens were only used to determine Mandard scores in. They were not used for RNA-Seq analyses.

2.4. How would the RNA/RNAseq data transfer to protein expression – e.g. immunohistochemistry? For a promising biomarker, an easy application in clinical routine would make sense.

We would like to refer to point 2.1 for this answer as well. We agree with the reviewer that an immunohistochemical biomarker would help clinical implementation. We stained for NANOG, SOX2, and OCT4 and concluded that these results were likely inaccurate (**Figure R6**).

In addition, and in an effort to explain these results, we attempted to identify and correlate our mRNA markers with their protein products in an unpublished novel proteomics dataset. However, we did not succeed in detecting the predictive markers at the protein level. This appears to be an abundance issue, which corroborates and possibly explains the negative immunohistochemistry results.

REVIEWER 3

The manuscript describes developing a model that predicts the response of EACs to CRT. This reveals NANOG and SOX4 as key associated transcription factors. Perturbation of NANOG was shown to reduce post-CRT viability, indicating that NANOG could be a biomarker for outcome prediction. The study is clearly written and well-performed with elegant experiments.

3.1. Sometimes the moving between different cell lines makes the story somewhat fussy as this is not always well explained.

This point is well taken. Several reasons urged us to move between cell lines across the paper. One is that not all cell lines were equally amenable to knockdown, especially the primary lines. Also, for *in vivo* experiments the 081R line typically shows the most consistent growth and is the most responsive to radiation-based combination therapies. Furthermore, some cell lines (such as Flo1) are very responsive to control stimulations such as TGF- β . Reasons like this unfortunately result in a limited overlap. Having stated that, our paper does report on what is to our knowledge, the biggest panel of (primary) EAC cell lines thus far. Also, in experiments such as those shown in Fig. 3e, candidate genes were knocked down across 8 different cell lines.

We now indicate in the panels which cell lines were used to make this more transparent.

Another point that needs to be discussed is that the data do not reveal whether the recurrences are CRT induced or may have been already present (at very low levels) before the treatment. This should at least be discussed.

We thank the reviewer for this comment. In the Netherlands, neoadjuvant chemoradiation (nCRT) followed by surgery is standard of care for resectable esophageal cancer. However, to be eligible for this curative treatment, patients need to be free of metastases as determined by imaging (CT, PET-CT). Therefore, all patients in this cohort for which we report tumor response grade (which is done on the resection specimen) were free of metastases at this point. This is now mentioned in the Methods section. To prove whether the recurrence was *induced* by CRT, would require a surgery-alone group which is not available.

3.2. Figure S1a, due to the few cells shown, it is unclear that 031M lost mesenchymal morphology, whereas 007B retained it. It looks like 031M already had a more MSC appearance without IR. Quantify and explain better.

The reviewer raises an important issue: Does the baseline mesenchymal phenotype determine plasticity? Although we used representative images for the manuscript, we agree that it is hard for the readers to interpret the baseline mesenchymal phenotype of the primary cell lines. To better show the heterogeneity of the used cell line panel, we improved the PCA plot originally found in Figure S2a and labeled the data points with the plasticity ranking (**Figure R1a**). The figure is now presented in the revised manuscript as Figure S2a: the PCA plot shows 031M to be near 007B, 081R and OE19. These cell lines are characterized by high CDH1, CD24, and EpCAM protein expression (Figure 1F). We also included images of the full EAC cell line panel in the revised manuscript (**Figure R1b** and Figure S2b). From this data, we conclude that baseline mesenchymal state does *not* define plasticity. For instance, although the 031M cells have a slightly higher mesenchymal baseline state than 007B cells, the mesenchymal shift during chemoradiation is significantly higher than the baseline difference.

3.3. Fig1D What were the absolute differences between the cell lines? I suppose 031M already had a lot higher level of Vim?

Please note that this information is shown in Figure 1F for *all* cell lines. The lines in that figure start at the baseline (pre-CRT) values of the indicated markers. Indeed, 031M expresses more VIM than

does 007B but as for our response above, we would like to point out again that baseline expression of mesenchymal markers does not predict plasticity.

3.4. *Fig1. d, mention below fig 1e should be Fig 1f? otherwise, the text above does not make sense.*

We apologize for the error and have corrected the mistake.

3.5. *Fig S1b. The markers are very inconsistent between cell lines. This should be mentioned.*

We thank the reviewer for pointing this out. This is one of the main findings of the paper: the propensity to undergo EMT is not similar between cell lines. We consider this *heterogeneity* rather than *inconsistency*. Note that Figure S1b shows relative change in response to CRT. We have modified the text on this.

3.6. *Fig2d label the x-axis as metastasis recurrences.*

We apologize for the omission and have changed the label on the x-axis in Figure 2d.

3.7. *Fig 2e, this is unclear. How can be distinguished between CRT-induced and already present metastasis? Why in the lung and not in other places? Is it related to ZEB1?*

See also our response with 3.1: EAC patients that are to receive neoadjuvant chemoradiation (nCRT) followed by surgery, are always assessed by imaging before the first treatment to ascertain no metastases are present. If metastases are found, nCRT with curative intent will be changed to palliative treatment. Therefore, all patients in the tumor response analysis have no indication of metastases (cM0) at the start of nCRT.

For EAC, the most common metastatic site is the liver, followed by lung, bone and brain (11). We observed the largest number (N=6) of metastases to the liver, in agreement with literature. However, we indeed observed the highest ZEB1 prediction score for metastases to the lung and bone. This can be explained by the highest ZEB1 prediction score representing the highest mesenchymal plasticity, and therefore the ability to seed to, and survive in, less likely metastatic niches such as the lung and bone. We mention this in the revised manuscript.

3.8. *Fig S4e, why suddenly 289B cells here? They were already low in confluency. The drug seems to have little effect on them. How about the other cell lines?*

We think the quality of the previously included images may have caused this misunderstanding. It is our opinion that the addition of drug to the CRT has an additional effect (compare 2nd to 3rd panel taken from that Figure):

Please note that quantitative data on confluence for all the cell lines is shown in Figure S4c.

3.9. *FigS4f can related sentence can be removed as it does not mean anything?*

We agree with the reviewer that Figure S4f is redundant including the related sentence. Both are now removed from the current manuscript.

3.10. *With FigS5a,b - both compounds? I see only NIC.*

We apologize for the error and have corrected the mistake.

3.11. *Top line 1 of page 12 Fig 4d, eE? Capital E can be removed?*

We have corrected the mistake.

REFERENCES

1. Pastushenko I, Brisebarre A, Sifrim A, Fioramonti M, Revenco T, Boumahdi S, et al. Identification of the tumour transition states occurring during EMT. *Nature*. 2018;556(7702).
2. Brown MS, Abdollahi B, Wilkins OM, Lu H, Chakraborty P, Ognjenovic NB, et al. Phenotypic heterogeneity driven by plasticity of the intermediate EMT state governs disease progression and metastasis in breast cancer. *Sci Adv*. 2022 Aug 5;8(31).
3. Eichelberger L, Saini M, Moreno HD, Klein C, Bartsch JM, Falcone M, et al. Maintenance of epithelial traits and resistance to mesenchymal reprogramming promote proliferation in metastatic breast cancer. *bioRxiv*. 2020 Apr 2;2020.03.19.998823.
4. Dings MPG, Zalm AP Van Der, Bootsma SJ, Waasdorp C, Liu D, Bailey P, et al. Estrogen related receptor alpha drives mitochondrial biogenesis and resistance to neoadjuvant chemoradiation in esophageal cancer. *Cell Reports Med*. 2022;Accepted.
5. Ebbing EA, Van Der Zalm AP, Steins A, Creemers A, Hermsen S, Rentenaar R, et al. Stromal-derived interleukin 6 drives epithelial-to-mesenchymal transition and therapy resistance in esophageal adenocarcinoma. *Proc Natl Acad Sci U S A*. 2019;116(6):2237–42.
6. Ebbing EA, Steins A, Fessler E, Stathi P, Lesterhuis WJ, Krishnadath KK, et al. Esophageal Adenocarcinoma Cells and Xenograft Tumors Exposed to Erb-b2 Receptor Tyrosine Kinase 2 and 3 Inhibitors Activate Transforming Growth Factor Beta Signaling, Which Induces Epithelial to Mesenchymal Transition. *Gastroenterology*. 2017 Jul 1;153(1):63-76.e14.
7. Steins A, Ebbing EA, Creemers A, van der Zalm AP, Jibodh RA, Waasdorp C, et al. Chemoradiation induces epithelial-to-mesenchymal transition in esophageal adenocarcinoma. *Int J Cancer*. 2019 Nov 15;145(10):2792–803.
8. Tarin D. The Fallacy of Epithelial Mesenchymal Transition in Neoplasia. *Cancer Res*. 2005 Jul 15;65(14):5996–6001.
9. Santamaria PG, Moreno-Bueno G, Portillo F, Cano A. EMT: Present and future in clinical oncology. *Mol Oncol*. 2017 Jul 1;11(7):718–38.
10. Moffitt RA, Marayati R, Flate EL, Volmar KE, Loeza SGH, Hoadley KA, et al. Virtual microdissection identifies distinct tumor- and stroma-specific subtypes of pancreatic ductal adenocarcinoma. *Nat Genet*. 2015 Sep 29;47(10):1168–78.
11. Ai D, Zhu H, Ren W, Chen Y, Liu Q, Deng J, et al. Patterns of distant organ metastases in esophageal cancer: a population-based study. *J Thorac Dis*. 2017 Sep 1;9(9):3023–30.

REVIEWERS' COMMENTS:

Reviewer #1 (Remarks to the Author):

The authors have now addressed my comments adequately.

Reviewer #2 (Remarks to the Author):

The authors have sufficiently addressed to my comments and as far as I can see the comments of the other reviewers as well. Thank you

Reviewer #3 (Remarks to the Author):

No further comments

RESPONSE TO REVIEWERS – Title: *The pluripotency factor NANOG contributes to mesenchymal plasticity and is predictive for outcome in esophageal adenocarcinoma*, with manuscript number **COMMSMED-23-0309A** Revision R2

REVIEWER 1

The authors have now addressed my comments adequately.

We are happy to hear we have adequately addressed all comments and thank the reviewer for improving the quality of the paper.

REVIEWER 2

The authors have sufficiently addressed to my comments and as far as I can see the comments of the other reviewers as well. Thank you

We thank the reviewer for acknowledging the added experiments and improving the quality of the manuscript.

REVIEWER 3

No further comments

We are happy to hear that no further adjustments are required to optimize the quality of the paper.